# Defense-Related Enzyme Activities and Metabolomic Analysis Reveal Differentially Accumulated Metabolites and Response Pathways for Sheath Blight Resistance in Rice

**DOI:** 10.3390/plants13243554

**Published:** 2024-12-19

**Authors:** Xiurong Yang, Shuangyong Yan, Yuejiao Li, Guangsheng Li, Yujiao Zhao, Shuqin Sun, Jingping Su, Zhongqiu Cui, Jianfei Huo, Yue Sun, Heng Yi, Zhibin Li, Shengjun Wang

**Affiliations:** 1Institute of Plant Protection, Tianjin Academy of Agricultural Sciences, Tianjin 300381, China; 2Tianjin Key Laboratory of Crop Genetics and Breeding, Institute of Crop Research, Tianjin Academy of Agricultural Sciences, Tianjin 300381, China; 3College of Agronomy & Resources and Environment, Tianjin Agricultural University, Tianjin 300384, China

**Keywords:** rice sheath blight, rice cultivar, defense enzyme, disease resistance, untargeted metabolomics

## Abstract

Rice sheath blight (RSB), caused by the pathogenic fungus *Rhizoctonia solani*, poses a significant threat to global food security. The defense mechanisms employed by rice against RSB are not well understood. In our study, we analyzed the interactions between rice and *R. solani* by comparing the phenotypic changes, ROS content, and metabolite variations in both tolerant and susceptible rice varieties during the early stages of fungal infection. Notably, there were distinct phenotypic differences in the response to *R. solani* between the tolerant cultivar Zhengdao22 (ZD) and the susceptible cultivar Xinzhi No.1 (XZ). We observed that the activities of five defense-related enzymes in both tolerant and susceptible cultivars changed dynamically from 0 to 72 h post-infection with *R. solani*. In particular, the activities of superoxide dismutase and peroxidase were closely associated with resistance to RSB. Metabolomic analysis revealed 825 differentially accumulated metabolites (DAMs) between the tolerant and susceptible varieties, with 493 DAMs responding to *R. solani* infection. Among these, lipids and lipid-like molecules, organic oxygen compounds, phenylpropanoids and polyketides, organoheterocyclic compounds, and organic acids and their derivatives were the most significantly enriched. One DAM, P-coumaraldehyde, which responded to *R. solani* infection, was found to effectively inhibit the growth of *R. solani*, *Magnaporthe grisea*, and *Ustilaginoidea virens*. Additionally, multiple metabolic pathways, including amino acid metabolism, carbohydrate metabolism, metabolism of cofactors and vitamins, and metabolism of terpenoids and polyketides, are likely involved in RSB resistance. Our research provides valuable insights into the molecular mechanisms underlying the interaction between rice and *R. solani*.

## 1. Introduction

Rice sheath blight (RSB), caused by the pathogen *Rhizoctonia solani*, predominantly manifests from the late tillering stage to the panicle differentiation stage of rice cultivation. This pathogen not only targets the leaf sheath, but also extends its invasion to the leaves and panicles of rice, severely compromising both the yield and quality of rice [1,2]. The proliferation of semi-dwarf high-yielding rice varieties, coupled with the increased application of nitrogen fertilizers and higher planting densities, has exacerbated the prevalence of RSB. Consequently, this has led to annual yield losses ranging from 10 to 30%, with particularly severe cases experiencing reductions as high as 50% [2,3,4,5]. To date, the primary strategy for RSB management has relied heavily on the application of chemical fungicides [5,6]. However, the excessive reliance on these fungicides has resulted in the emergence of drug-resistant strains, a decline in rice quality, and significant environmental pollution. Moreover, there is a dearth of rice varieties that exhibit inherent resistance or immunity to RSB. This underscores the critical need to intensify research efforts aimed at elucidating the molecular mechanisms underlying rice resistance to RSB [7].

Plants have evolved a sophisticated array of defense mechanisms to recognize and combat pathogen invasion. A pivotal component of the initial immune response is the rapid generation of reactive oxygen species (ROS), which is a hallmark of pattern-triggered immunity (PTI) [8,9]. This ROS burst serves as the frontline of defense during the early stages of pathogen encounter. ROS, including hydrogen peroxide (H_2_O_2_), superoxide, and hydroxyl radicals, are reactive forms of oxygen and are toxic products. H_2_O_2_ is the most stable among ROS, and the levels can serve as an indicator of overall ROS content. H_2_O_2_ can bind to the DAB (3,3′-diaminobenzidine) dye, resulting in the accumulation of a brown material, as evidenced by microscopic examination [10]. Typically, plants meticulously regulate the equilibrium between ROS production and scavenging throughout their interaction with pathogens, thereby triggering a cascade of downstream defense reactions [11]. Enzymatic scavengers of ROS, such as superoxide dismutase (SOD), peroxidase (POD), and catalase (CAT), are integral to modulating ROS levels and are closely linked with the plant’s capacity for disease resistance [12,13]. For instance, the rice resistant line IRBB4 exhibited a markedly reduced ROS accumulation compared to the susceptible line IR24 at 48 h post-inoculation with *R. solani*. This differential response is critical, as unchecked ROS accumulation can lead to plant cell death, thereby facilitating pathogen growth and colonization [14]. Furthermore, ROS activates the phenylalanine ammonia lyase (PAL)-mediated biosynthesis of defensive lignin and phenolic compounds. These compounds are instrumental in reinforcing the plant’s defense arsenal [15,16]. Polyphenol oxidase (PPO) is another crucial enzyme that catalyzes the production of phenolic compounds, which serve as a bulwark against the incursion of plant pathogens [17]. Additionally, POD has been identified as a key player in sustaining the ROS-redox equilibrium, promoting the regeneration and fortification of the plant cell wall [18].

When plants encounter pathogen invasion, they activate critical metabolic pathways, reflecting the biochemical shifts that transpire during plant–pathogen interactions. Our previous research has delineated that lipids, lipid-like molecules, phenylpropanoids, polyketides, organoheterocyclic compounds, organic acids and their derivatives, lignans, neolignans, and related compounds are significantly enriched in the panicle blast-resistant rice cultivar jinsui18 following inoculation with the rice blast fungus, *Magnaporthe oryzae* [19]. Comprehensive KEGG enrichment analysis indicates that the phenylpropanoid pathway is crucial for disease resistance in rice sheath blight (RSB)-resistant cultivars [20,21]. Two recent metabolomics studies have reported remarkable changes in photosynthesis and sugar metabolism in rice in response to *R. solani* infection. Furthermore, reactive oxygen species (ROS), salicylic acid (SA), jasmonic acid (JA), aromatic aliphatic amino acids, and phenylpropanoid intermediates accumulate during the infection stage, consistent with the response typically observed during lesion formation by necrotrophic pathogens [7].

The molecular mechanisms underpinning RSB have been extensively studied, yet the isolation and identification of a specific RSB resistance gene have eluded researchers. A more profound understanding of the molecular interactions between *R. solani* and its rice host is pivotal for the development of enhanced resistance in rice varieties. Previously, we employed an inoculation-based screening method to identify two rice varieties exhibiting markedly distinct levels of resistance to RSB: Zhengdao 22 (ZD), which demonstrates a high tolerance to the disease, and XinZhi No.1 (XZ), which is highly susceptible. Furthermore, we conducted a comparative transcriptomic analysis between these two cultivars, uncovering both positive and negative regulators involved in the response to *R. solani* infection [19]. To enhance our understanding of the interactions between rice cultivars and *R. solani*, this study investigated disease progression, levels of ROS, and metabolite variations in these tolerant and susceptible rice varieties during the early stages of *R. solani* infection. Additionally, it explored the metabolite response to this fungal pathogen. Interestingly, in our preliminary research, we found that the metabolite P-coumaraldehyde showed a response to the invasion of sheath blight fungus. P-coumaraldehyde stands out as a critical component involved in the reinforcement of plant cell walls. Specifically, it plays a role in the biosynthesis of lignin-like materials, which are deposited in the cell walls of cucumber (*Cucumis sativus*) and squash (*Cucurbita* spp.) upon induction by pathogens or elicitors [22]. However, the role of this metabolite in response to rice RSB has yet to be reported. The investigation of these parameters is expected to yield a detailed and integrated perspective on the biochemical mechanisms that confer resistance to RSB in rice. The findings from this research will contribute to a more holistic view of the complex biochemical processes that are engaged in the resistance to RSB. This knowledge is fundamental for guiding future breeding programs and genetic engineering initiatives, aiming to cultivate rice varieties with robust resistance to this significant agricultural challenge.

## 2. Results and Discussion

### 2.1. Phenoty Ences in Response to R. solani Infection Between Tolerant and Susceptible Rice Cultivars

The tolerant rice cultivar ZD and the highly susceptible cultivar XZ were inoculated with the rice sheath blight pathogen, *Rhizoctonia solani*, to evaluate their phenotypic responses. The resistance of the two cultivars to rice sheath blight was assessed at the heading stage. Phenotypic responses to *R. solani* infection were subsequently observed and compared from 0 to 72 h post-inoculation (hpi). Dynamic phenotypic changes, indicative of their resistance levels, were observed in both the tolerant and susceptible cultivars between 0 to 72 hpi with *R. solani*. At 48 hpi, the tolerant variety ZD exhibited green–yellow spots on the leaf sheaths, whereas typical cloudy spots became apparent at 72 hpi. In contrast, the highly susceptible cultivar XZ showed green–yellow spots at 24 hpi, with more pronounced cloudy spots evident by 48 hpi (Figure 1A). Accordingly, the onset of symptoms was significantly delayed in the tolerant variety ZD compared to the susceptible cultivar XZ. Additionally, the lesion length was significantly greater in the susceptible cultivar XZ compared to the tolerant variety ZD, as depicted in Figure 1B. Both the tolerant cultivar ZD and the susceptible cultivar XZ underwent significant phenotypic changes post-inoculation, with ZD demonstrating a delayed onset of symptom development compared to XZ.

The tolerant rice cultivar ZD and the highly susceptible cultivar XZ were inoculated with the rice sheath blight pathogen *R. solani*. Phenotypic responses to *R. solani* infection were observed and compared between the ZD and XZ cultivars, with assessments made at intervals ranging from 0 to 72 hpi. Their resistance to rice sheath blight was assessed during the heading stage.

Dynamic phenotypic changes were observed in both the tolerant and susceptible cultivars from 0 to 72 hpi. By 48 hpi, the tolerant variety ZD exhibited green–yellow spots on the leaf sheaths, with typical cloudy spots becoming evident by 72 hpi. In contrast, the highly susceptible cultivar XZ showed green–yellow spots at 24 hpi, with the appearance of more pronounced cloudy spots by 48 hpi (Figure 1A). Consequently, symptom onset was delayed in the tolerant variety ZD compared to the susceptible cultivar XZ. Additionally, lesion lengths were significantly longer in the susceptible cultivar XZ compared to the tolerant variety ZD, as depicted in Figure 1B. Both the tolerant ZD and susceptible XZ cultivars exhibited significant phenotypic changes following inoculation, with ZD demonstrating a delayed onset of symptom development compared to XZ.

Further investigation was conducted to examine the differences in lesion development between the two rice varieties at 72 hpi. On the surface of the tolerant ZD variety, the mycelium showed reduced growth and a lower frequency of infestation mats. In comparison, the susceptible XZ variety had dense mycelium on the leaf sheaths, with numerous infestation mats observed (Figure 2). These findings suggest that tolerant varieties can inhibit fungal growth and the formation of infestation mats, thereby limiting the rapid spread of disease symptoms.

The resistance to rice sheath blight at the heading stage was characterized for both tolerant and susceptible rice varieties, with the results detailed in Table 1. The variety ZD exhibited superior resistance to rice sheath blight (RSB), with a lesion extension ratio of 0.17 ± 0.01 mm relative to plant height at the heading stage, indicating a moderately resistant phenotype. In contrast, the susceptible variety XZ showed a lesion extension ratio of 0.78 ± 0.02 mm, which, compared to ZD, was evaluated as highly susceptible. These results were consistent with a previous observation that the main difference between resistant and susceptible rice cultivars was the timing of responses to *R. solani* infection [21].

### 2.2. ROS Response to R. solani Between Tolerant and Susceptible Rice Cultivars

The dynamic changes in ROS levels in both tolerant and susceptible rice varieties were monitored after inoculation, ranging from 0 to 72 hpi. ROS levels were observed to increase over time in both varieties inoculated with *R. solani*. There was a gradual increase in ROS levels, and the tolerant variety ZD exhibited minimal brown precipitate in the leaf sheath at 24 hpi, indicating extremely low ROS levels in the cells at that time. In contrast, in the susceptible variety XZ, obvious brown precipitation was evident in the leaf sheath at the same time, indicating an accumulation of ROS (Figure 3). Both the tolerant and susceptible cultivars exhibited a gradual increase in intracellular ROS levels from 0 to 72 hpi, yet the levels in the tolerant cultivar ZD consistently remained lower than those in the susceptible cultivar XZ.

Plants have evolved a variety of defense mechanisms, including intracellular immune receptors and surface-localized pattern recognition receptors (PRRs), to detect pathogen-associated molecular patterns (PAMPs) and initiate the immune response against invading pathogens. Among these responses, the rapid production of reactive oxygen species (ROS) is a pivotal regulatory element in plant immunity, functioning both as a signaling molecule for defense gene activation and a toxic agent against pathogens. However, ROS can also be exploited by pathogens to facilitate their invasion, growth, and development. In our experiments, a time-dependent sustained elevation of ROS levels was observed in both tolerant and susceptible rice cultivars post-inoculation. Notably, the tolerant variety ZD consistently exhibited approximately 50% lower ROS levels compared to the susceptible variety XZ at the peak of the immune response, underscoring the importance of efficient ROS degradation in rice resistance to sheath blight. Our findings corroborate those of Oreiro et al., who reported a significantly lower accumulation of ROS in the rice resistant line IRBB4 compared with the susceptible line IR24 at 48 hpi with *R. solani*, highlighting the role of ROS management in disease resistance [14].

### 2.3. Dynamic Changes in Defense Enzyme Activities in Tolerant and Susceptible Rice Cultivars Following R. solani Infection

To further assess the response of tolerant and susceptible rice to *R. solani* infection, we examined the activity changes in five defense-related enzymes, namely superoxide dismutase (SOD), catalase (CAT), peroxidase (POD), polyphenol oxidase (PPO), and phenylalanine ammonia-lyase (PAL), from 0 to 72 hpi (Figure 4). Dynamic changes in the activities of these enzymes were evident in both resistant and susceptible cultivars (Figure 4).

In the tolerant cultivar ZD, SOD activities increased from 0 to 72 hpi, and the susceptible cultivar XZ exhibited a similar pattern. However, at all time points measured, the SOD activity was consistently higher in the tolerant ZD compared to the susceptible cultivar XZ (Figure 4A). In the tolerant cultivar ZD, CAT activities initially decreased from 0 hpi to reach a minimum at 24 hpi, subsequently peaking at 48 hpi before declining once more. In contrast, the susceptible cultivar XZ showed a distinct trend, with its activities peaking at 48 hpi and then declining (Figure 4B). In the tolerant cultivar ZD, POD activities peaked at 48 hpi and then decreased. Conversely, in the susceptible cultivar XZ, activities decreased to a minimum at 24 hpi, increased to a peak at 48 hpi, and subsequently declined (Figure 4C).

A critical component of the antioxidative system is the equilibrium between the generation and scavenging of ROS. Enzymes involved in ROS scavenging play a role in modulating ROS levels and sustaining resistance to pathogens [23]. The activity of these enzymes is closely associated with plant disease resistance. For instance, a resistant faba bean (*Vicia faba*) cultivar exhibited higher enzymatic ROS-scavenging capacity during *Botrytis fabae* infection than a susceptible one [24]. Resistance to rice panicle blast correlates with the activities of ROS-scavenging enzymes [25]. In our current study, we measured the activities of three ROS-scavenging enzymes-SOD, POD, and CAT-early after inoculation. We demonstrated that ROS levels in the tolerant cultivar ZD remained consistently lower than those in the susceptible cultivar XZ, indicating that the tolerant rice cultivar ZD exhibited superior ROS-scavenging capacity. The activity of the ROS-scavenging enzyme SOD increased continuously from 0 to 72 hpi, and was consistently higher in the tolerant cultivar ZD compared to the susceptible XZ, which is consistent with the trend of ROS levels after inoculation in both cultivars. This suggests that SOD plays an important role in RSB resistance due to its ROS scavenging capacity. The rice genome contains seven genes encoding for SOD enzymes, which are crucial for various stress responses [26]. Four SOD-encoding genes, OsCSD3, OsSOD1, OsMSD, and OsFSD2, are known to play important roles in response to abiotic stress [27,28,29,30]. However, the role of these genes in rice disease resistance, particularly in RSB, has yet to be elucidated and warrants further investigation.

The activity of polyphenol oxidase (PPO) exhibited significant variation between the tolerant and susceptible rice varieties. In the tolerant variety ZD, PPO activity rapidly increased from 0 to its peak at 24 hpi, after which it gradually decreased. In contrast, the susceptible cultivar XZ showed an opposite trend, with activity initially rising slightly from 0 to 24 hpi, then falling to a minimum at 48 hpi, and subsequently increasing rapidly. The activity of PPO was significantly higher in the tolerant variety ZD than in the susceptible XZ across the entire time course (Figure 4D).

PPO, a copper-containing phenolase enzyme, is ubiquitously present in microorganisms, animals, and plants, where it plays a crucial role in various physiological processes. It can modify proteins by catalyzing reactions with various compounds, including amino and phenolic groups, leading to alkylation and thus influencing protein function and fate. PPO contributes to the organism’s defense mechanisms against biotic stresses, including resistance to pathogenic infections and insect attacks [31]. Significantly, in the present experiment, PPO activity was higher in the tolerant cultivar ZD than in the susceptible cultivar XZ upon *R. solani* infection. Notably, PPO activity in the tolerant cultivar ZD rapidly increased during the early stages post-inoculation. The findings indicate that PPO plays a role in combating pathogens, and the level of rice resistance to RSB is correlated with the level of PPO activity. The rice genome encodes two PPO genes, *Os01g0793300* and *Os04g0624500*, as annotated in the KEGG database. The PPO-encoding gene *Os04g0624500* (*Phr1*) is responsible for the phenol reaction phenotype and plays a role in positive selection during rice domestication [32]. However, the role of PPO-encoding genes in rice disease resistance has not been fully revealed. According to the results of this study, the activity of PPO is likely closely associated with rice RSB resistance, and revealing the molecular mechanism of its action is of great significance.

In the tolerant ZD, PAL activity decreased from 0 to 72 hpi. In contrast, the susceptible cultivar XZ exhibited an initial increase in activity from 0 to 24 hpi, followed by a decrease from 24 to 72 hpi. Overall, PAL enzyme activity demonstrated a gradual decline from 0 to 72 hpi with *R. solani* in both the tolerant and susceptible varieties (Figure 4E).

### 2.4. Metabolic Profiling via Ultra-Performance LC-MS Analysis

Metabolomic analysis was conducted on leaf samples collected from tolerant and susceptible rice cultivars at 48 hpi. As detailed in Table 2, the quantity of metabolites present in each sample was ascertained. Mass spectrometry was used to compare analyte peaks with commercial reference standards, leading to the identification of 3391 metabolites, including 1394 metabolites in negative mode and 1997 in positive mode (Figure 5A,B and Appendix A). Among the identified metabolites, 2890 were consistently detected across both the tolerant and susceptible cultivars, as well as in both infected and non-infected samples. A total of 3367 metabolites were detected in the infected samples, while 3299 were found in the non-infected control samples. Moreover, 3227 metabolites were detected in samples from both the tolerant and susceptible cultivars (Figure 5A,B). The presented data provide valuable evidence regarding the differences in the response to *R. solani* infection between the tolerant cultivar ZD and the susceptible XZ.

### 2.5. Metabolomic Differences Between Tolerant and Susceptible Rice Cultivars in Response to R. solani

Principal component analysis (PCA) of metabolites among the four samples revealed a distinct separation between the tolerant ZD and susceptible cultivar XZ, with the first two components accounting for 38.9% of the variance. These findings reveal a larger discrepancy in metabolic activity between the tolerant and susceptible cultivars, as compared with the differences observed between infected and non-infected samples (Figure 5C). The PCA results indicate that the majority of the variance in the metabolite profiles could be attributed to the genetic differences between the cultivars.

Upon constructing the OPLS-DA models to investigate the differences between tolerant and susceptible rice cultivars, a distinct clustering of the cultivars under both infected and non-infected conditions was observed (Figure 5D–F). Furthermore, permutation tests validated the OPLS-DA models, as indicated by the plots that clearly distinguished between tolerant and susceptible cultivars in both infected and non-infected samples.

Moreover, the heatmap-generated clusters of differential metabolites clearly segregated the tolerant from the susceptible cultivars, reflecting significant differences in their metabolome compositions (Figure 6). Differential metabolites were categorized into two distinct groups based on their accumulation patterns: Group 1 metabolites were highly accumulated in the susceptible cultivar XZ and less so in the tolerant variety ZD, while Group 2 metabolites were found at high levels in the tolerant cultivar and were less abundant in the susceptible one. These findings further corroborate the significant disparities between tolerant and susceptible varieties in terms of their metabolomes.

Following the initial metabolomic profiling, subsequent orthogonal partial least squares-discriminant analysis (OPLS-DA) of the comparisons among the four samples identified 825 differentially accumulated metabolites (DAMs). These were characterized by high variable importance in projection (VIP) values > 1 and statistically significant Student’s *t*-test *p*-values < 0.05 (Figure 7 and Appendix A). In the comparison between ZD_WK1 and ZD_WK2, representing the metabolic differences between infected and non-infected samples of the tolerant cultivar ZD, 33 out of 80 differentially accumulated metabolites (DAMs) were unique. In the comparative metabolomic analysis of the susceptible cultivar XZ, 37 out of 107 differentially abundant metabolites (DAMs) were uniquely present in the infected sample XZ_WK1 compared to the non-infected sample XZ_WK2, suggesting a clear metabolic divergence in response to infection. Among the differentially abundant metabolites (DAMs), P-coumaraldehyde was notably present at elevated levels in infected samples of both the tolerant and susceptible cultivars, underscoring a shared metabolic response to infection when comparing infected and non-infected samples (Appendix A).

In the comparative metabolomic analysis between the non-infected samples XZ_WK2 and ZD_WK2, 416 DAMs were identified. Among these, 187 DAMs were found to be unique to this specific comparison, suggesting distinct metabolic profiles between the two non-infected conditions. This finding indicates a distinct metabolic profile between the tolerant cultivar ZD and the susceptible cultivar XZ, independent of infection by *R. solani* (Appendix A). In the comparative metabolomic analysis between the tolerant cultivar ZD (XZ_WK1) and the susceptible cultivar (ZD_WK1), both infected by *R. solani*, we identified 493 DAMs. Among these, 240 were unique to this comparison. Notably, 263 were significantly lower in abundance, whereas 230 were markedly higher in the tolerant cultivar ZD (Figure 7 and Appendix A). The top 30 DAMs are depicted in Figure 8 and detailed in Appendix A. These include eleven phenylpropanoids–polyketides, such as six flavonoid glycosides, which are integral to plant growth, development, and defense mechanisms, including antimicrobial activity and disease prevention. Additionally, the list comprises eight organoheterocyclic compounds, four organic oxygen compounds, two lipids–lipid-like molecules, two organic acids–derivatives, and one lignan–neolignan and related compound.

### 2.6. Analysis of the Inhibitory Effect of P-Coumaraldehyde Against Pathogens

P-coumaraldehyde, a cinnamic aldehyde and a member of the phenylpropanoids, was present at high levels in infected samples of both the tolerant and susceptible cultivars, suggesting a potential role in the response to pathogen infection. This suggests that P-coumaraldehyde may have significant importance for rice in its ability to cope with pathogen invasion. A stress tolerance assay was performed to determine the inhibitory effect of P-coumaraldehyde on various pathogens. As shown in Figure 9 and Table 3, P-coumaraldehyde has been demonstrated to possess inhibitory properties against *R. solani*, *Magnaporthe grisea*, and *Ustilaginoidea virens*. P-coumaraldehyde is integral to plant defense, contributing to rapid lignification and the synthesis of defensive compounds through the phenylpropanoid pathway. Under stress, this pathway’s up-regulation increases lignin precursors, strengthening cell walls against pathogens [22]. This is consistent with our metabolomic analysis results, which showed that the levels of this metabolite increased in both susceptible and resistant varieties following inoculation with *R. solani*. This stress tolerance assay of P-coumaraldehyde may be more intriguing because it suggests that the metabolite might act directly on the pathogen, rather than through enhancing the plant’s defensive barriers, since the identification process did not involve any participation from the plant; this finding aligns with earlier reported results [33].

### 2.7. DAMs Related to RSB Resistance

In total, 493 DAMs were observed between tolerant and susceptible cultivars infected at 48 hpi, with 467 of these being classified according to the super class on the HMDB 4.0 database (Figure 10A and Appendix A). Lipids and lipid-like molecules, including phenol lipids (60), fatty acyls (36), and steroids–steroid derivatives (12), were the most frequent DAMs, accounting for 24.2% of the total 467 classified DAMs (113 in number). This was followed by organic oxygen compounds (21.2%, 79), phenylpropanoids and polyketides (16.5%, 77), organoheterocyclic compounds (14.9%, 70), organic acids and derivatives (13.06%, 61), benzenoids (8.77%, 41), fewer nucleosides, nucleotides and analogues (11), lignans, neolignans and related compounds (7), alkaloids and derivatives (3), and hydrocarbons (2), each representing a percentage of the total 467 classified DAMs.

Previous research has demonstrated that pathogen-induced metabolites significantly contribute to the enhanced resistance of plants against various pathogens. In this study, we identified 493 distinct defense-associated metabolites in response to *R. solani* infection, including lipids and lipid-like molecules, organic oxygen compounds, phenylpropanoids, and polyketides. These metabolites are also enriched following inoculation with the rice blast fungus *M. oryzae* [25].

Lipids are one of the major components of plant cell biological membranes, forming an essential barrier that separates the cell from its external environment and prevents microbial invasion [34]. Recent studies have highlighted the critical role of lipids in plant defense responses. These lipids are categorized into eight classes based on the hydrophobic and hydrophilic regions of their chemical backbones: fatty acyls, glycerolipids, glycerophospholipids, sphingolipids, sterol lipids, phenol lipids, saccharolipids, and polyketides [34]. In the present study, we observed that DAMs, belonging to phenol lipids, fatty acyls, and steroid derivatives, were the predominant lipids differentially accumulated in response to *R. solani* infection. Most phenolic lipids among the DAMs are terpenoid compounds, including monoterpenoids, diterpenoids, sesquiterpenoids, and triterpenoids. Some of these may exhibit fungicidal activity against *R. solani* or other fungi, given that numerous antimicrobial substances have been isolated from terpenoids [35]. Fatty acids are crucial for forming surface barriers against abiotic and biotic stresses, maintaining membrane integrity, and providing energy for various metabolic processes [36]. Our findings indicate that phenol lipids and fatty acids are likely to play an important role in responding to the *R. solani* challenge. However, further investigation is necessary to elucidate the precise mechanisms underlying the defense-promoting properties of these metabolites.

In our results, 55 differentially abundant metabolites (DAMs) of organic oxygen compounds were identified as belonging to the subclass of carbohydrates and carbohydrate conjugates. These compounds may serve as nutrient sources during *R. solani* infection. Successful pathogen invasion also hinges on their capacity to exploit the nutrient sources, such as carbohydrates and carbohydrate conjugates, provided by plants [37]. Consequently, plants may evolve countermeasures to counteract the effects of invasive pathogens. A potential strategy for controlling rice blast disease, which involves modulating the carbon allocation in the host plant, has been proposed [38]. Employing a similar approach to modulate carbon allocation could be instrumental in assessing resistance to rice sheath blight.

### 2.8. Metabolic Pathways of Related to Rice RSB Resistance

Further analysis of the 493 DAMs associated with metabolic pathways revealed that multiple pathways might be responsible for the metabolic changes in rice cultivars when challenged with *R. solani*. Figure 10B illustrates that metabolic pathways with more than 10 DAMs encompass the biosynthesis of other secondary metabolites (15 DAMs), amino acid metabolism (15 DAMs), carbohydrate metabolism (13 DAMs), and metabolism of cofactors and vitamins (11 DAMs). These metabolic pathways may be associated with resistance to rice sheath blight. Among these, metabolites associated with amino acid metabolism and the metabolism of cofactors and vitamins also exhibited significant differences between infected and non-infected samples in the tolerant cultivar ZD (Figure 10C).

Critical metabolic pathways are activated in plants upon infection by pathogens, a response crucial for their defense mechanisms [39]. Elucidating these pathways could provide valuable insights and inform the development of specific strategies for enhancing plant resistance systems. Our current study revealed that the biosynthesis of secondary metabolites, amino acid metabolism, carbohydrate metabolism, and the metabolism of cofactors and vitamins were found to be associated with resistance to RSB. The secondary metabolites produced by plant pathogenic fungi, such as those causing rice RSB, are complex and possess unique structures that play a significant role in the pathogen–host interaction. *R. solani*, the causal agent of RSB, is known to produce four types of secondary metabolites: melanin, tenuazonic acid, nectriapyrones, and pyriculols [40]. Many of these metabolites participate in the interaction between *R. solani* and rice, yet the specific metabolites produced during this interaction often remain elusive, highlighting a significant gap in our understanding of the disease resistance mechanism [40].

Pathogens’ successful infection relies on their ability to exploit the nutrient sources available from plants. Additionally, their capacity to penetrate and overcome the host plants’ defensive mechanisms is critical [37]. Primary metabolites, including specific specialized metabolites, are essential for plant development, growth, and reproduction, and also protect against biotic and abiotic stresses [41]. Amino acid metabolism has been identified as having a substantial impact on the growth, asexual reproduction, differentiation, and virulence of pathogens [10,42,43,44]. Amino acids are involved in nitrogen assimilation processes and act as precursors for the synthesis of various defense-related secondary metabolites [45].

## 3. Materials and Methods

### 3.1. Plant Materials and Growth Conditions

In this study, two japonica rice cultivars, resistant cultivar Zhengdao22 (ZD) and susceptible cultivar Xinzhi No.1 (XZ) were obtained from the Tianjin Academy of Agricultural Sciences. Rice were cultivated in the netting house at the WuQing Experimental Base in Tianjin, with a row pitch of 20 cm and a row space of 20 cm. The soil of the experimental fields was loam (pH 7.1–8.0) and was fertilized with 15 kg/ha nitrogen before experiment. Water deficiency was supplemented when needed.

### 3.2. R. solani Strains, Culture Conditions, and Inoculation

The pathogen *R. solani* TJ-1 was provided by the institute of Plant Protection of Tianjin Academy of Agricultural Sciences. The culture conditions of the pathogen were as previously reported by Yang et al. [19]. The isolates TJ-1 were grown on potato sucrose agar (PSA) medium for 3 to 4 days at 28 °C. The method of inoculation was carried out according to Zhu et al. [4]. Briefly, wooden tips after sterilization were placed on the PSA plates with *R. solani* and then cultured at 28 °C for 3 days. When these tips were covered with *R. solani*, they were inserted slightly into the second sheath of rice seedlings. Sterile tips without inoculum were used as the control. In total, twenty plants of each cultivar were injected, with three replicates measured. The inoculated plants were evaluated for disease response as a percentage of the relative lesion height (RLH%) 4 weeks after inoculation, as shown below:$$RLH\%=\frac{Lesionheight\;(cm)}{Plantheight\;(cm)}$$

Rice sheath blight resistance was measured using the 0–9 scale of the Standard Evaluation System (SES) for rice (IRRI, 2002) [46].

Samples were collected and recorded at 0, 24, 48, and 72 h after inoculation, each including three replicates. Leaves were cut from each plant above the leaf sheath in infected and non-infected resistant and susceptible cultivar groups per treatment at each time point. Samples per variety were pooled and immediately frozen in liquid nitrogen and stored at −80 °C. For each treatment, three independent replicates were included.

### 3.3. Colonization Observation of R. solani

The colonization status of *R. solani* was observed under a microscope via Trypan blue staining as described in a previous work [47]. Rice leaf sheaths collected were firstly immersed in 50% ethanol for 3 days, then transferred to 2.5%KOH and bathed at 90 °C for 30 min. The root samples were dipped in 1% HCl overnight and then washed with water three times. They were then stained with 0.05% Trypan blue (using acid glycerol as solvent) bathed at 90 °C for 20 min, and immersed in acid glycerol for decolorating for 1 to 2 h. The samples were mounted on glass sides and then observed and recorded under a Nikon ECLIPSE NI-U research microscope (Nikon, Tokyo, Japan).

### 3.4. Enzyme Activity Determination

Leaf samples (100 mg) were homogenized in ice-cold buffer (pH 7.8) and the supernatant after centrifuging was used as enzyme extract for further analysis. The spectrophotometry analyses were then conducted on an SP-756P spectrophotometer (Shanghai Spectrum Instruments Co., Ltd., Shanghai, China). SOD activity was measured via a SOD-1-W detection kit (Suzhou Comin Biotechnology Co., Ltd., Suzhou, China). POD activity was assayed using a POD-1-Y detection kit (Suzhou Comin Biotechnology Co., Ltd.), while CAT activity was determined using a CAT-1-Y detection kit (Suzhou Comin Biotechnology Co., Ltd.) by measuring H_2_O_2_ decomposition as the decrease in absorbance at 240 nm. PAL activity was measured via a PAL-1-Y detection kit (Suzhou Comin Biotechnology Co., Ltd., Suzhou, China). PPO activity was measured via a PPO-1-Y detection kit (Suzhou Comin Biotechnology Co., Ltd., Suzhou, China). PAL activity in the extract were measured at a wavelength of 490 nm and PPO activity was assessed following oxidation of catechol at 420 nm.

### 3.5. Stress Tolerance Assay

For the stress tolerance assay, the ager disks containing mycelium of *Rhizoctonia solani*, *Magnaporthe grisea,* and *Ustilaginoidea virens*, were inoculated to PSA plants supplemented with 0, 0.1, 1, 10, and 100 μg/mL of P-coumaraldehyde (Shanghai yuanye Bio-Technology Co., Ltd., Shanghai, China; 1000 μg/mL stock solution in chloroform). PSA plants supplemented with sterile ddH_2_O were used as control. Each treatment consisted of 4 replicates. The colony diameter of *R. solani* was recorded and measured after culturing for 48 h at 25 °C, and 5 days after inoculated for *M. grisea* and *U. virens*. The least squares method was employed to formulate a linear regression equation, specifically a toxicity regression equation of P-coumaraldehyde against three pathogens [48]. The half-maximal effective concentration (*EC*_50_) value was determined based on the toxicity regression equation and was calculated to compare the toxicity.

### 3.6. Untargeted Metabolomic Profiling

Three rice leaves of each treatment were harvested and flash-frozen in liquid nitrogen immediately after collection. Samples were lyophilized and then stored at −80 °C. The extraction solvent containing methanol/water (4:1, *v*:*v*) which used 0.02 mg/mL internal standard (L-2-chlorophenylalanine) was added to each freeze-dried sample to obtain the leaves’ metabolites. After grinding for 6 min at 50 Hz in a frozen tissue grinder at −10 °C, the samples were sonicated for 30 min at −20 °C. Then the mixture was centrifuged at 14,000 rpm for 15 min at 4 °C. The collected supernatant was used for ultrahigh-performance liquid chromatography-mass spectrometry (UHPLC-MS/MS) analysis.

The Thermo UHPLC system, equipped with an ACQUITY UPLC HSS T3 column (100 mm × 2.1 mm i.d., 1.8 µm; Waters, Milford, MA, USA), was utilized to separate the metabolites with mobile phases A (95% water/5% acetonitrile plus 0.1% formic acid), and B (47.5% acetonitrile/47.5% isopropanol/5% water plus 0.1% formic acid). The sample volume injected was 5 µL, and the flow rate was maintained at 0.3 mL/min. The column temperature was held constant at 40 °C, and all samples were kept at 4 °C throughout the analysis. The mass spectrometric data were gathered using either a positive or negative ion mode on a Thermo UHPLC-Q Exactive HF-X mass spectrometer with an ESI source. The capillary voltage and spray shield were set to 3500 V and the sheath gas low rate and nebulizer gas low rate were set to 50 arb and 13 arb, respectively, with a capillary temperature of 325 °C. Spectra were acquired between *m*/*z* 70 and 1050, with collision energy in MS/MS mode set to 20–40–60 V.

### 3.7. Data Analysis

All data from this experiment were expressed as the mean ± SE of three biological replicates. Statistical analysis was performed by using a one-way analysis of variance (ANOVA) with Duncan’s multiple range test at a 5% significance level.

The data were pre-processed with Progenesis QI 2.3 (Nonlinear Dynamics, Waters, Milford, MA, USA). By accounting for the accurate mass, the MS/MS fragment spectra, and the isotope ratio difference, reliable biochemical databases such as the Human Metabolome Database (HMDB) (https://www.hmdb.ca/, accessed on 29 September 2022), the Metlin database (https://metlin.scripps.edu/, accessed on 29 September 2022), Lipidmaps (https://www.lipidmaps.org/, accessed on 29 September 2022), and KEGG Compound database (https://www.kegg.jp/kegg/compound/, accessed on 29 September 2022) were used to identify the mass spectra of these metabolites. On the Majorbio Cloud Platform (https://cloud.majorbio.com, accessed on 29 September 2022), a multivariate statistical analysis was conducted using the R package ropls (Version 1.6.2, https://bioconductor.org/packages/release/bioc/html/ropls.html, accessed on 29 September 2022). The metabolites between different groups were distinguished via orthogonal partial least squares-discriminant analysis (OPLS-DA). In the OPLS-DA model, the variable importance in the projection (VIP) was determined. *p* values were calculated using the single-dimensional paired Student’s *t* test. Based on VIP values > 1 and *p* values < 0.05, differentially expressed metabolites (DAMs) were detected. Based on database searches (https://www.genome.jp/kegg/, accessed on 29 September 2022), the DAMs were mapped onto Kyoto Encyclopedia of Genes and Genomes (KEGG) pathways.

## 4. Conclusions

In this study, we systematically compared the differences in phenotypes, defense-related enzymes activities, and metabolomes between a tolerant and a susceptible rice variety in response to *R. solani* infection at an early stage. Our research has shown that there are significant phenotypic differences between tolerant and susceptible cultivars following pathogen invasion, accompanied by differences in ROS scavenging enzymes, particularly SOD, and PPO, suggesting that defense-related enzymes are involved in resistance to RSB. We identified 825 differentially accumulated metabolites (DAMs) between RSB-tolerant and susceptible cultivars that may be involved in resistance to *R. solani* infection. Among these, P-coumaraldehyde has been directly shown to possess antimicrobial properties. We identified 493 differentially abundant metabolites, especially lipids and lipid-like molecules, organic oxygen compounds, as well as phenylpropanoids and polyketides, which are notably associated with resistance against RSB. In addition, the pathways for biosynthesis of other secondary metabolites, amino acids, carbohydrates, and cofactors and vitamins are also linked to RSB resistance. Such research outcomes are likely to be beneficial in the optimization of resistance to RSB, in addition to providing elementary data relevant to future plant-disease prevention.

## Figures and Tables

**Figure 1 plants-13-03554-f001:**
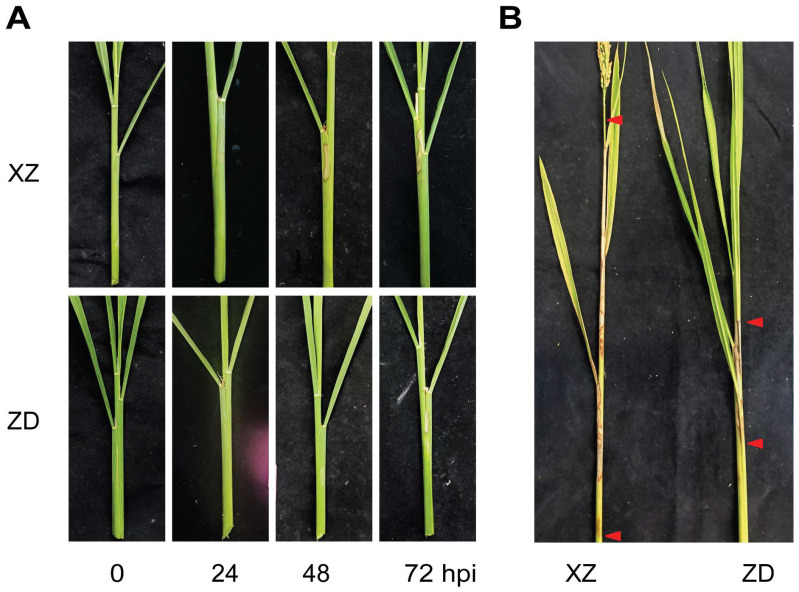
Phenotypes of tolerant and susceptible cultivars against *R. solani*. Phenotype changes in susceptible cultivar Xinzhi No.1 (XZ) and resistant cultivar Zhengdao22 (ZD) from 0–72 hpi (**A**) and full heading stage (**B**) after inoculation are shown. The red triangle shows the length of the lesion.

**Figure 2 plants-13-03554-f002:**
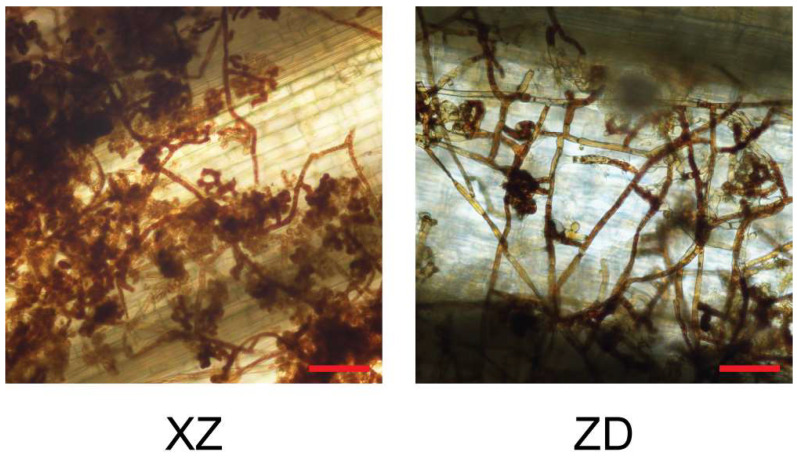
Difference of infection morphology in the surface of leaf sheath of susceptible cultivar Xinzhi No.1 (XZ) and resistant cultivar Zhengdao22 (ZD) inoculated after 72 h. Scale bar = 200 μm.

**Figure 3 plants-13-03554-f003:**
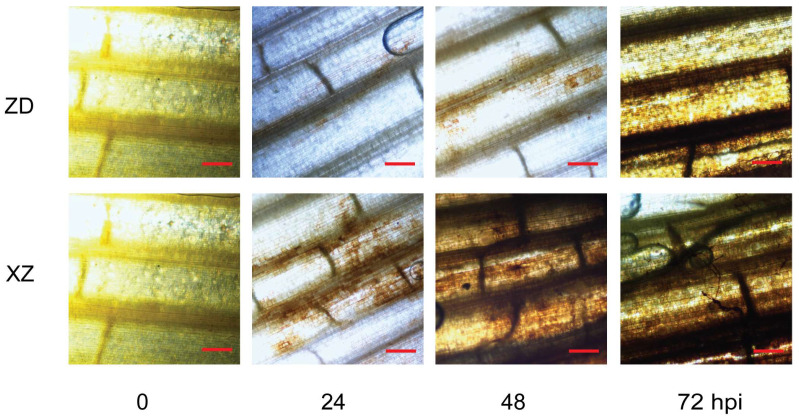
Detection of H_2_O_2_ in rice leaf sheath using DAB (3,3′-diaminobenzidine) from 0 to 72 hpi by *R solani*. Scale bar = 200 μm.

**Figure 4 plants-13-03554-f004:**
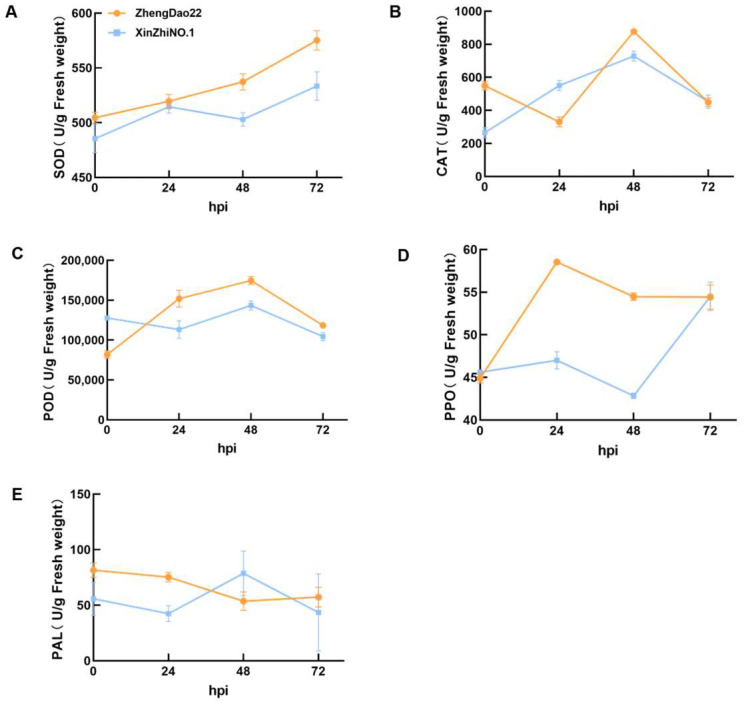
Dynamic changes in the activities of defense enzymes SOD (**A**), POD (**B**), PPO (**C**), PAL (**D**), CAT (**E**) in resistant cultivar Zhengdao22 and the susceptible cultivar Xinzhi No.1 from 0 to 72 hpi.

**Figure 5 plants-13-03554-f005:**
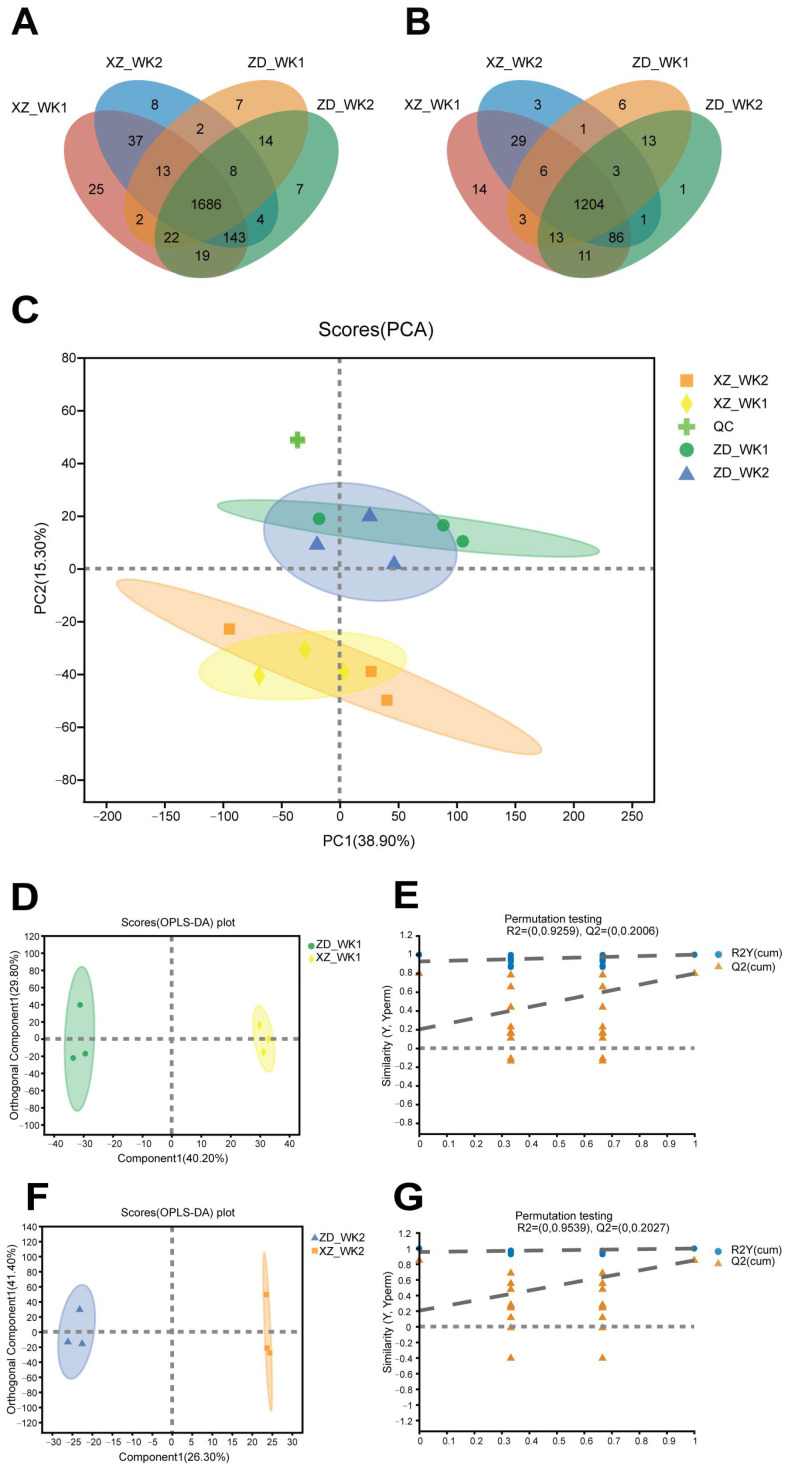
Metabolite profiling data in tolerant and susceptible varieties. Venn diagram of metabolic profiling in tolerant and susceptible varieties in positive mode (**A**) and negative mode (**B**). ZD_WK1 is the metabolome of tolerant cultivar ZD infected with *R. solani* sampled at 48 hpi. ZD_WK2 is the metabolome of tolerant cultivar ZD without infection, sampled at 48 hpi XZ_WK1 is the metabolome of susceptible cultivar XZ infected with *R. solani* sampled at 48 hpi. XZ_WK2 is the metabolome of resistant cultivar XZ without infection sampled at 48 hpi. (**C**) PCA scores of metabolomes in rice leaves for infected and non-infected tolerant and susceptible cultivars. OPLS-DA scores plots (**D**) and permutation tests (**E**) for metabolite profiling of tolerant cultivar ZD and susceptible cultivar XZ with *R. solani* infection. OPLS-DA scores plots (**F**) and permutation tests (**G**) for metabolite profiling of tolerant cultivar ZD and susceptible cultivar XZ without *R. solani* infection.

**Figure 6 plants-13-03554-f006:**
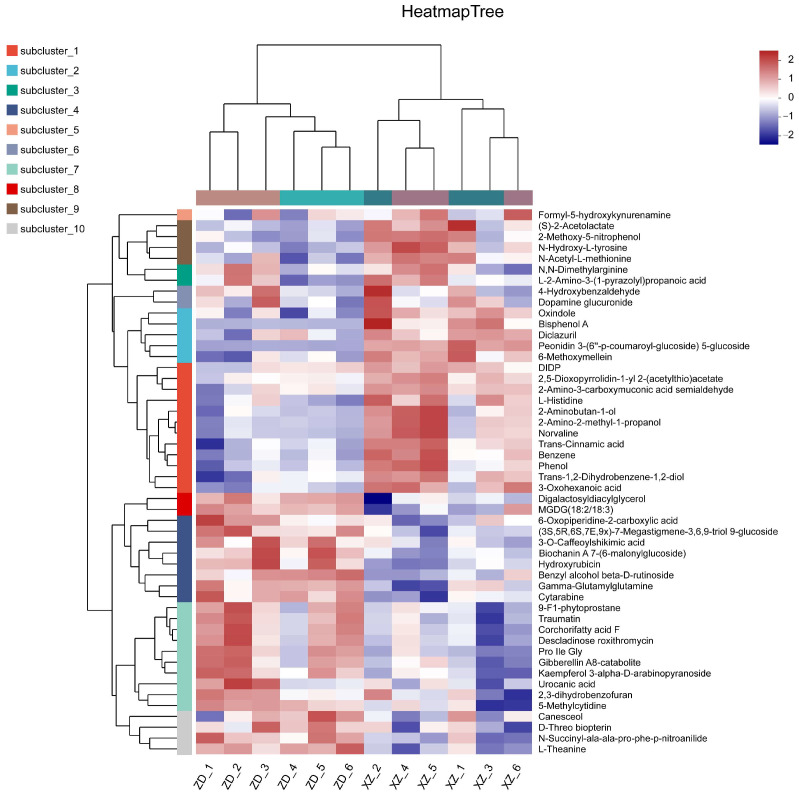
Heatmap analysis of representative differential metabolites in two groups. XZ_1 and XZ_2, XZ_3 are the metabolome of susceptible cultivar XZ infected with *R. solani* sampled at 48 hpi with three repeats; XZ_4, XZ_5, and XZ_6 are the metabolome of susceptible cultivar XZ without infection sampled at the same time as XZ_1 to XZ_3; ZD_1 to ZD_3 are the metabolome of tolerant cultivar ZD infected with *R. solani* sampled at 11 hpi with three repeats. ZD_4, ZD_5, and ZD_6 are the metabolome of tolerant cultivar ZD without infection sampled at the same time as ZD_1 to ZD_3. group1, metabolites with low abundance in tolerant cultivar ZD and high abundance in susceptible cultivar XZ; group2, metabolites with high abundance in tolerant cultivar ZD and low abundance in susceptible cultivar XZ.

**Figure 7 plants-13-03554-f007:**
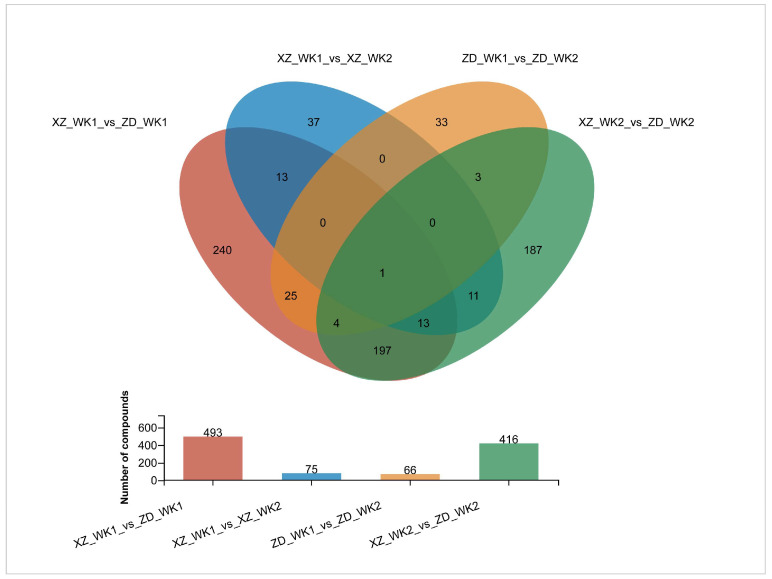
Venn diagram of differential metabolites in four groups. XZ_WK1_vs_ZD_WK1 represents differential metabolites between susceptible cultivar XZ and tolerant cultivar ZD infected with *R. solani*; XZ_WK1_vs_XZ_DW2 represents differential metabolites between infected and non-infected samples of susceptible cultivar XZ;ZD_WK1_vs_ZD_WK2 represents differential metabolites between infected and non-infected samples of tolerant cultivar ZD; XZ_WK2_vs_XZ_WK2 represents differential metabolites between susceptible cultivar XZ and tolerant cultivar ZD without infection.

**Figure 8 plants-13-03554-f008:**
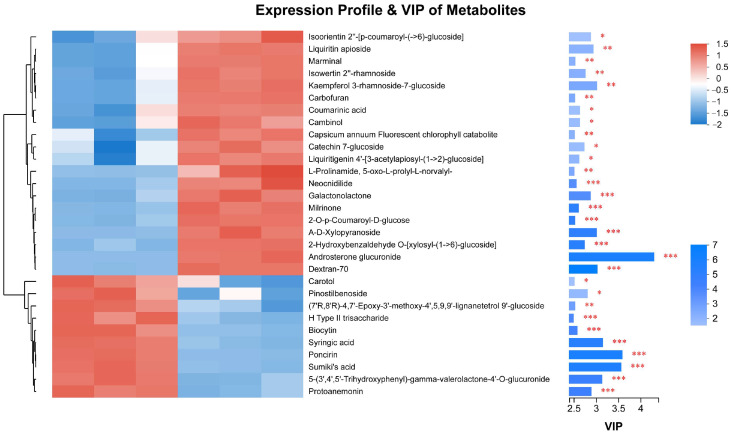
Expression profiling of the top 30 differential metabolites of tolerant and susceptible cultivars with *R. solani* infection. Relative abundances of the top 30 differential metabolites between tolerant and susceptible cultivars with *R. solani* infection. Data are normalized against unit variance. Comparisons were generated via hierarchical cluster analysis using an average linkage method based on Euclidian distance. Shades from blue to red represent increasing metabolite levels (** p* < 0.05, *** p* < 0.01, **** p* < 0.001 for groups compared using one-way ANOVA).

**Figure 9 plants-13-03554-f009:**
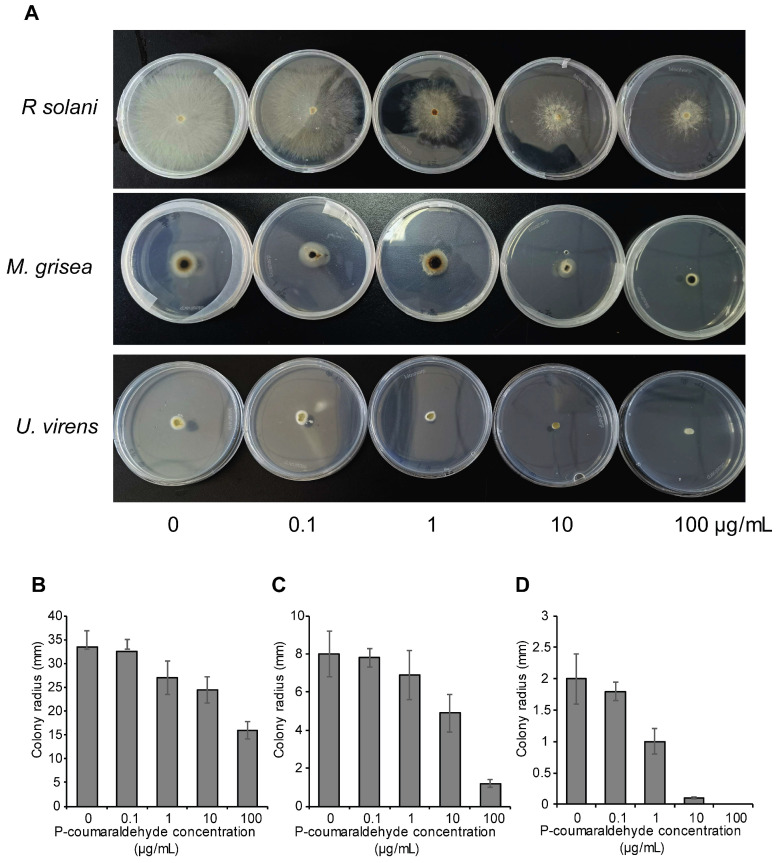
Stress tolerance assay of P-coumaraldehyde against pathogens. (**A**) Inhibitory effect of P-coumaraldehyde against *Rhizoctonia. solani*, *Magnaporthe grisea*, and *Ustilaginoidea virens*. Pathogens were incubated on PSA medium with 0, 0.1, 1, 10, and 100 μg/mL of P-coumaraldehyde. The colony diameter of *R. solani* (**B**) was measured after culturing for 48 h and 5 days after inoculated for *M. grisea* (**C**) and *U. virens* (**D**).

**Figure 10 plants-13-03554-f010:**
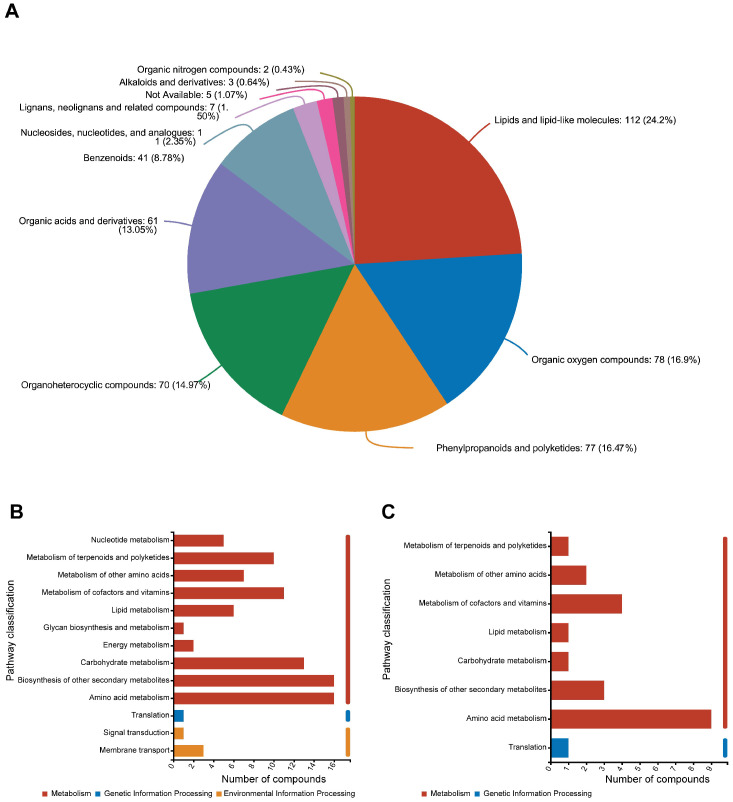
(**A**) Classification of differential metabolites of tolerant and susceptible cultivars with *R. solani* infection. (**B**) Pathway classification of differential metabolites between tolerant cultivar ZD and susceptible cultivar XZ with *R. solani* infection. (**C**) Pathway classification of differential metabolites between infected and non-infected samples for tolerant cultivar ZD.

**Table 1 plants-13-03554-t001:** The resistance to RSB of susceptible cultivar Xinzhi No.1 (XZ) and resistant cultivar Zhengdao22 (ZD).

Variety	Spot Length/mm	Plant Height/mm	Ratio	Resistance
XZ	545.0 ± 18.0	698.0 ± 20.9	0.78 ± 0.02	High susceptible
ZD	140.0 ± 8.7	818.0 ± 20.2	0.17 ± 0.01	Moderately resistant

**Table 2 plants-13-03554-t002:** Metabolic profiling results.

Samples	Total Metabolites	Descriptions
ZD_WK1	3003	Infected ZD sampled at 48 hpi
ZD_WK2	3235	Non-infected ZD control sampled at 48 hpi
XZ_WK1	3313	Infected XZ sampled at 48 hpi
XZ_Wk2	3234	Non-infected XZ control sampled at 48 hpi

**Table 3 plants-13-03554-t003:** Toxicity determination of P-coumaraldehyde against three pathogens.

Target	Correlation Coefficient	Toxicity Regression Equation	EC_50_ (mg/L)	Confidence Interval (95%)
*Rhizoctonia solani*	1.0000	Y = 3.7287 + 0.6589X	85.2409	17.81–408.04
*Magnaporthe grisea*	0.9903	Y = 3.8223 + 1.2721X	12.7890	3.64–44.94
*Ustilaginoidea virens*	0.9974	Y = 5.1211 + 1.4632X	0.8265	0.11–6.13

## Data Availability

The data presented in this study are available in the article and Appendix A here. The raw metabolomics data generated in this study have been submitted to the Open Archive for Miscellaneous Data in China National Center for Bioinformation (https://ngdc.cncb.ac.cn/omix/) accessed on 31 December 2024 under accession number OMIX002710.

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
