# Peer review of "Defense-Related Enzyme Activities and Metabolomic Analysis Reveal Differentially Accumulated Metabolites and Response Pathways for Sheath Blight Resistance in Rice"

_plants, 2024, doi:10.3390/plants13243554_

Round 1
Reviewer 1 Report
Comments and Suggestions for Authors
Dear Authors,
I have really enjoyed you MS.
Rice is an important food staple in the world, thus the effect of the pathogenic fungus examination is also very important.
1. Introduction
The introduction need to expand, because the 2.6 paragraph is about P-coumaraldehyde, but the introduction is not contain anything about it. Please expand the introduction with that compound. However, the mention paragraph is contain results about the inhibitory effect on pathogenic species but missing the introduction and materials and methods which connected to them.
2. Results and discussion
Overall, the results are not enough separated.
2.1. paragraph
The ZD and XZ cultivars are missing from the introduction. This part is clear results without discussion. Please expand this.
2.2 paragraph
Please check the whole part, because for example the “Reactive oxygen species (ROS), including hydrogen peroxide (H2O2), superoxide, and hydroxyl radicals, are reactive forms of oxygen and are toxic products [23]. H2O2 is the most stable among ROS, hence, the levels of H2O2 were determined in this study, and the levels can serve as an indicator of overall ROS content [23].” May better in the introduction.
DAB (3,3′-diaminobenzidine), please check the abbreviations.
The Oreiro et al is missing reference, please check.
2.3. paragraph
Figure 4. Not easy to find the different between the shapes. Change it.
2.4-5 paragraph
Why did you measured the 48 hpi? Is there any relevance that chose this treatment?
Figure 5. What is the different between the 5A and 5B? Why is not used ZD and XZ named cultivars and used R391 and R405? It is hard to find out what is what.
Table S1. From this table it is clear the figure 5A is the positive mode, but have you checked the measured components? During the identification there are several false information. If you do not check the components one-by-one there could be a lot of missed peaks. For example: “metab_15410: (-)-2-Difluoromethylornithine” is an inhibitor of polyamine biosynthesis, rice is not going to synthetize. HMDB (Human Metabolite DataBase) contain Human metabolites and not the plant metabolites, but it is able to find which species were measured. Not only the 2-Difluoromethylornithine is the problem, I have found carbofuran metab_10955 and this compound was in the figure 8. This is a very toxic pesticide.
Please check the components one-by-one.
2.6 paragraph
As I mentioned earlier this part has no previous information. Why is this part is here? At the material and method there is no quantification of the LS MS part, but here there is quantified information.
2.7. paragraph
493 DAMs have found, 465 DAMs classified? Why? Please explain me why is not possible to classify the component. The classification is only depend on the structural of the molecule. After the measurement the structural is known, in this case the classification is clear.
2.8.paragraph
It has to rewrite base on the mentioned points.
3. Materials and methods
3.2 paragraph
There is a missing references, like IRRI 2002, and the hpi is not need to write here.
3.4 paragraph
Yang et al is not contain the enzyme measurement. Check it.
3.5 paragraph
PDA or PSA? Earlier you wrote about PSA. Check it.
3.6 paragraph
The used mobile phase is written twice, check it.
4. Conlcusion
Based on the mentioned points please write it again.
Best regards
Reviewer 2 Report
Comments and Suggestions for Authors
This manuscript offers a detailed overview of rice sheath blight (RSB) and its causal agent, Rhizoctonia solani, supported by well-chosen references that highlight the importance of the study. While the introduction is comprehensive, it could be improved with a clearer summary of the gaps in current research and a more direct explanation of how this study aims to fill them. The research design is strong, utilizing comparative phenotypic, enzymatic, and metabolomic analyses to distinguish between tolerant and susceptible rice cultivars. However, 48 hours post-inoculation as a key time point could be better justified to reinforce the study's rationale.
The methods section provides sufficient detail for reproducibility, but specifying the controls used in enzymatic assays and metabolomic profiling would eliminate potential ambiguity. The results are presented clearly, with well-organized figures, tables, and statistical analyses. Integrating metabolomics with enzymatic activity data is a standout feature of the work. That said, some figures, particularly the heatmaps, would benefit from more detailed legends to aid interpretation.
The findings support the conclusions well, highlighting critical insights such as the role of p-coumaraldehyde in pathogen resistance and the involvement of phenylpropanoids and polyketides. Expanding on the practical implications of these discoveries for rice breeding programs would enhance the study's overall impact. While the manuscript is well-written and accessible, there are minor grammatical issues and occasional awkward phrasing, especially in the abstract and results sections, that should be polished to improve clarity and readability.
